



# Simulating the evolution of the topography-climate coupled system

Kyungrock Paik[1] and Won Kim[1]

[1] School of Civil, Environmental, and Architectural Engineering, Korea University
145 Anam-ro, Seongbuk-gu, Seoul, 02841, South Korea

**Correspondence:** Kyungrock Paik (paik@korea.ac.kr)

**Abstract.** Landscape evolution models simulate the long-term variation of topography under given rainfall scenarios. In reality, local rainfall is largely affected by topography, implying that surface topography and local climate evolve together. Herein, we develop a numerical simulation model for the evolution of the topography-climate coupled system. We investigate how simulated topography and rain field vary between 'no-feedback' and 'co-evolution' simulations. Co-evolution simulations

produced results significantly different from those of no-feedback simulations, as illustrated by transects and time evolution in rainfall excess among others. We show that the evolving system keeps climatic and geomorphic footprints in asymmetric transects and local relief. We investigate the roles of the wind speed and the time lags between hydrometeor formation and rainfall (called the delay time) in the co-evolution. While effects of the wind speed and delay time were thought to compensate each other in the evolving morphology, we demonstrate that the evolution of the coupled system can be more complicated than

previously thought. The channel concavity on the windward side becomes lower as the imposed wind speed or the delay time grows. This tendency is explained with the effect of generated spatial rainfall distribution on the area-runoff relationship.

## 1 Introduction

Mutual influences between topography and climate have been widely recognized (e.g., Molnar and England, 1990; Masek et al., 1994; Willett, 1999; Roe et al., 2008). The spatial distribution of precipitation leads the spatial variability in surface runoff

and erosion rate (e.g., Reiners et al., 2003; Moon et al., 2011; Bookhagen and Strecker, 2012; Ferrier et al., 2013), affecting landscape formation in the long term (e.g., Anders et al., 2008; Colberg and Anders, 2014). On the other hand, topography greatly affects precipitation distribution. Encountering upslope terrain on the path of moist air movement, wind raises an air parcel to a higher altitude, leading to cooling, condensation, and formation of hydrometeors (Smith, 1979). Such orographic effect gives rise to the pattern in precipitation fields correlated with topography, which has been reported around the world

(Puvaneswaran and Smithson, 1991; Park and Singh, 1996; Bookhagen and Burbank, 2006; Falvey and Garreaud, 2007, among others). The interaction between surface topography and precipitation field has a far-reaching implication, i.e., landscape and local climate evolve together as a single entity. Roe et al. (2008) suggested an analytical framework to describe the feedbacks among orographic precipitation, fluvial erosion, and critical wedge orogeny. Field-based evidences of such co-evolution have also been reported (e.g., Norton and Schlunegger, 2011; Champagnac et al., 2012). Nevertheless, the co-evolutionary dynamics

and the resultant patterns in topography as well as local climate have been largely unexplored.



Mathematical modeling can be a sensible approach for the deeper investigation of feedbacks and controls in the evolutionary dynamics of the topography-climate coupled system. This requires a sophisticated numerical simulation model which links various surface processes and local rainfall generation processes. Since the 1990s, whole landscape evolution models have been popularly utilized for quantitative analysis and theoretical understanding of terrestrial processes (e.g., Willgoose et al., 1991a;

Howard, 1994; Densmore et al., 1998; Zhang et al., 2016). Readers may refer to Temme et al. (2017) for a comprehensive review on mathematical models. Most early simulation studies assumed spatially uniform rainfall. Some studies have imposed spatially varying overland flow and evaluated its contribution to landform evolution (e.g., Pelletier, 2009; Wobus et al., 2010). Han et al. (2014) considered simple spatial rainfall distributions in simulating landform evolution of the island of Hawaíi, using the landscape evolution model CHILD (Tucker et al., 2001). Ward and Galewsky (2014) studied the impact of spatial rainfall

distribution on stream incision using a 1D model. While these studies adopted spatial variability in rainfall or runoff, their spatial distribution was treated invariant over time, ignoring topographic feedbacks on the precipitation field.

Co-evolution modeling with the consideration of topographic feedbacks was initiated by Masek et al. (1994) who coupled a simple orographic rainfall model with the cellular-automata surface process model of Chase (1992). Braun and Sambridge (1997) simulated the co-evolution of topography and local rain field with their landscape evolution model CASCADE. They

adopted a simple orographic rainfall function where the rainfall depth becomes proportional to the surface elevation. Anders et al. (2008) incorporated the physically-based orographic rainfall model of Smith and Barstad (2004) (referred to as the SB model in this paper) into the CASCADE model. Adopting a similar approach, Han et al. (2015) coupled the SB model with the CHILD model and investigated the imprint of orographic rainfall on the steady-state morphology focusing on longitudinal profiles and river network organization. It was suggested that rainfall gradients perpendicular to mountain range produce narrower

catchments. Colberg and Anders (2014) adopted a simple Gaussian function for the spatial rainfall distribution wherein the local surface elevation is used as a variable. They ran co-evolution simulations by imposing the rainfall function into the CASCADE model and claimed that the location of precipitation maxima can determine whether the passive margin escarpment retreats and becomes gentler or steepens over time.

Despite aforementioned progresses, co-evolution modeling is a relatively new area and many open questions remain. In

this study, we aimed to advance our understanding on two fundamental subjects: (1) topographic changes that co-evolutionary dynamics bring in and (2) the respective role of meteorological variables on co-evolution. First, we explored how the relief and transect of topography evolve under the co-evolutionary setting. In this context, we investigated the feedbacks among local climate, surface processes, and the given tectonic uplift condition. Particularly, we questioned how the channel profile evolves under the co-evolutionary setting. The effect of orographic rainfall on the channel profile was investigated by Roe et al. (2002)

for a 1D single corridor. For the river network within a landscape, Han et al. (2015) discussed the effect of orographic rainfall on channel concavity, as "high rainfall rates at the ridge top lead to mainstem channels that have relatively low concavity." We sought a framework, which generalizes this description, from theoretical investigation in the present study.

Second, we investigated how meteorological variables control co-evolution. The most up-to-date strategy for simulating co-evolution can be found in the studies of Anders et al. (2008) and Han et al. (2015). In those studies, a rain field is contin-

uously updated with evolving terrain by coupling the SB and landscape evolution models, and therefore, feedbacks between





topography and climate were simulated. The SB model generates the rainfall field with many meteorological variables such as the wind vector $\boldsymbol{v}$, the conversion time $t_c$ from cloud water to hydrometeors, and the fallout time of hydrometeors $t_f$. The sum of the latter two quantities is referred to as the total delay time $t_d$ (Smith and Evans, 2007), i.e., $t_d = t_c + t_f$. A greater $\|\boldsymbol{v}\|$ or $t_d$ locates rainfall further downward. In this sense, their combined effects on an orographic rain field can be given as the

non-dimensional delay time (or non-dimensional cloud drift time) (Barstad and Smith, 2005), i.e.,

$$t_* = \|\boldsymbol{v}\| t_d / L \tag{1}$$

where the length scale $L$ is given as the mountain half-width. Considering the long-term effect of an orographic rain field on the topography formation, Anders et al. (2008) claimed that $t_*$ "has a profound impact on topography (and precipitation patterns)." However, we wonder whether $t_*$ can represent the effect of local climate on long-term topography evolution. While both $\|\boldsymbol{v}\|$

and $t_d$ contribute horizontal displacement of rain field, they may play oppositely in producing overall rainfall amount. Considering these notions, the evolution of orographic rain field in conjunction with evolving topography may be more complicated than previously thought.

We aimed to address these questions via improving our understanding on the theoretical dynamics of the co-evolutionary system. We approached the proposed agenda with mathematical simulations. We investigated a hypothetical landscape and

local climate converging a quasi-steady state. The rest of this paper is organized in the order written below. We present the proposed co-evolution model structure in Section 2. Numerical simulation results along with scientific insights are presented in Section 3. In-depth discussions about implications of simulation results and current model limitations are given in Section 4. Finally, conclusions are presented in Section 5.

## 2    Coupling Orographic Rainfall Generation and Topography Evolution

Modelling orographically induced precipitation is complex and can be attempted in various ways depending on the level of detailed physics incorporated (Barros and Lettenmaier, 1994). Among those, vertically integrated analytical models provide a reasonable balance between simulation accuracy and model complexity. In these models, the vertically-averaged wind is expressed as the advection wind vector $\boldsymbol{v}$:

$$\boldsymbol{v} = U\boldsymbol{i} + V\boldsymbol{j} \tag{2}$$

where $U$ and $V$ are wind speed components, and $\boldsymbol{i}$ and $\boldsymbol{j}$ are unit vectors along the $x$ and $y$ directions (we adopt the Cartesian coordinate system), respectively.

One of the well-known models of this type is the upslope model (e.g., Collier, 1975), which expresses cloud water $S$ at a coordinate $(x, y)$ in a raster domain as the inner product of $\boldsymbol{v}$ and the gradient $\nabla z$ of surface elevation field $z$ (Smith and Barstad, 2004):

$$S(x,y) = C_w \boldsymbol{v} \cdot \nabla z + S_o. \tag{3}$$





Note that both $\boldsymbol{v}$ and $\nabla z$ are vectors evaluated at the coordinate $(x, y)$. In equation (3), $S_o$ is the background non-orographic large-scale vertically integrated condensation rate. Here, $C_w$ is the thermodynamic uplift sensitivity factor, given as

$$C_w = \rho_\nu \Gamma_m / \gamma \tag{4}$$

where $\rho_\nu$, $\Gamma_m$, and $\gamma$ are the saturation vapor density, moist adiabatic lapse rate, and environmental lapse rate, respectively.

Assuming that condensed water immediately falls on the ground, the precipitation rate $P = S$.

Smith (2003) relaxed the assumption of immediate fallout by introducing the hydrometeor conversion time $t_c$ and fallout time $t_f$. Accordingly, the Fourier transform of $P(x, y)$ was presented as

$$\hat{P}(k, l) = \frac{\hat{S}(k, l)}{(1 + i\omega t_c)(1 + i\omega t_f)} \tag{5}$$

where $\hat{S}(k, l)$, $i$, and $\omega$ are the Fourier transform of $S(x, y)$, the imaginary unit ($\sqrt{-1}$), and the intrinsic frequency, respectively.

Here, $k$ and $l$ are the components of the horizontal wavenumber vector and $\omega = Uk + Vl$. Then, precipitation distribution can be obtained via its inverse transformation as

$$P(x, y) = \int \int \hat{P}(k, l) e^{i(kx + ly)} dk dl + P_o. \tag{6}$$

Here, $P_o$ is an optional term of the background precipitation rate, i.e., the rate without the presence of any orographic effect. In the absence of terrain, $P_o = S_o$.

Smith and Barstad (2004) improved the calculation of cloud water by considering airflow dynamics as

$$\hat{S}(k, l) = \frac{C_w i\omega \hat{z}(k, l)}{1 - iH_w m} \tag{7}$$

where $\hat{z}(k, l)$ is the Fourier transform of the elevation field $z$, $H_w$ is the moist layer height, and $m$ is the vertical wave number. $H_w$ is given as

$$H_w = -\frac{R_v T^2}{\Lambda \gamma} \tag{8}$$

where $R_v$, $T$, and $\Lambda$ are the gas constant for vapor (461 J/kg/K), the near-surface air temperature, and the latent heat of vaporization (2,501 kJ/kg at 273.15 K), respectively. Instead of keeping $\Lambda$ a constant, its variation with $T$ is considered in the proposed model, using

$$\Lambda = 1,918.46 \left(\frac{T}{T - 33.91}\right)^2 \text{ (in kJ/kg)} \tag{9}$$

which was suggested by Henderson-Sellers (1984). In the hydrostatic limit ($N_m^2 \gg \omega^2$), $m$ is given as

$$m(k, l) = N_m \sqrt{k^2 + l^2} \operatorname{sgn}(\omega) / \omega \tag{10}$$

where the effective moist static stability (Fraser et al., 1973) is approximated as

$$N_m = \sqrt{g(\gamma - \Gamma_m)/T}. \tag{11}$$





Here, $g$ is the gravitational acceleration. Considering the time lags (equation (5)) and airflow dynamics (equation (7)), a better estimation of the precipitation field is given (Smith and Barstad, 2004) as

$$\hat{P}(k,l) = \frac{C_w i\omega \hat{z}(k,l)}{(1 - iH_w m)(1 + i\omega t_c)(1 + i\omega t_f)} \tag{12}$$


which is the model mentioned earlier in this paper as the SB model. The orographic precipitation can be obtained from its inverse transformation (equation (6). Precipitation can be in various phases such as rain and snow. Our scope of precipitation phase is the rainfall at the time when the drop touches the ground.

The SB model is coupled with the surface process model LEGS (Landscape Evolution model using Global Search) (Paik,
2012). LEGS is a physically-based mathematical model implemented on the same raster domain as the SB model. $P$ calculated from the SB model is used to generate the overland flow. There is no evapotranspiration or infiltration in our simulations. In this sense, we name $P$ the rainfall excess rate. Then, the flow discharge per unit channel width $q$ is calculated for every cell considering its upslope area $A$ and $P$ given in the area.

GD8 (global eight-direction) method (Paik, 2008) was incorporated in LEGS for surface flow path extraction, which helps
obtain reasonable results in terms of evolutionary speed and characteristics of simulated topography. For this study, we revised and upgraded the source code of LEGS, adopting many improved features. One of them is the replacement of GD8 with the improved GD8 method (Shin and Paik, 2017), which resolves known technical issues of the original GD8. The improved GD8 requires the assignment of start cells in a digital elevation model. This requires sorting cells according to their elevation values. Merge sort algorithm, invented by John Von Neumann (see e.g., Knuth, 1987), is adopted for this task.

In a real landscape, transport-limited and detachment-limited conditions are mixed, where the latter often appears upstream. In theoretical modeling studies, however, the entire landscape has been assumed as one of these conditions. Such a simple assumption eases the interpretation of simulation results by excluding higher order complexities. The choice between the two conditions depends on the scope and purpose of each study. For example, Willgoose et al. (1991a) adopted the transport-limited condition. Earlier co-evolution studies of Anders et al. (2008) and Han et al. (2015) used the detachment-limited condition.
The dominance between transport-limited and detachment-limited can be related with timescales. Temme et al. (2017) noted that "the assumption that there is always a large supply of transportable material is more likely true when considering very long (Ma) timescales – but on smaller timescales, temporary lack of transportable material can occur." In this study, the simulations are run for a long term (5 Ma) initiated from a flat topography. Accordingly, a transport-limited alluvial landscape is considered.

The sediment transport rate per unit channel width $q_s$ (mass/time/length) is calculated for each cell along a flow path as a
function of $q$ and the local slope $\|\nabla z\|$, i.e.,

$$q_s = C\|\nabla z\|^\alpha q^\beta \tag{13}$$

where $C$ is an empirical coefficient. Equation (13) has been widely used in many landscape evolution studies (e.g., Smith and Bretherton, 1972; Willgoose et al., 1991a; Tucker and Slingerland, 1994; Braun and Sambridge, 1997), and its origin is described in the Appendix A. Shallow landslide is considered in the simple way, previously applied by Tucker and Slingerland
(1994); any slopes greater than the angle of repose $\Theta$ shall fail, translating upslope mass at the rate $q_f$ to downhill until the local





slope reaches $\Theta$. This contribution is combined with the fluvial sediment transport to form the total sediment flux $\xi(= q_s + q_f)$ at every cell. With the cell-to-cell mass transfer calculated above, the elevation field is updated over time $t$ on the basis of the mass conservation along the flow path $p$, given as

$$\frac{\partial z}{\partial t} = -\frac{1}{(1-\lambda)\rho_s}\frac{\partial \xi}{\partial p} + u \tag{14}$$

where $\lambda$, $\rho_s$, and $u$ are the porosity, density of a sediment particle, and tectonic uplift rate, respectively.

In the coupled model proposed in this study, the rain field generated by the SB model is fed into the LEGS model. Then, the updated surface topography obtained from the LEGS model becomes input to the SB model to generate the rain field for the next time step. The SB model involves the Fourier transform of the elevation field (equation (12)) and the inverse transform of the rain field (equation (6)). We implemented the full coupling between the LEGS and SB models, i.e., the topography and

local rain field feed into each other at every iteration. To lighten any computation load in these operations, Kim (2014) adopted the fast Fourier transform method (Cooley and Tukey, 1965) and developed an efficient code, which we utilized in this study.

## 3   Applications

We applied the developed model to theoretical cases of mountain range formation, uplifting from an initially (almost) flat surface at the sea level. Simulations were implemented on a square domain composed of $200 \times 200$ cells, wherein each cell

has an area of $0.5 \times 0.5$ km². In numerical simulations, spatial and temporal resolutions should be related to each other. For the selected calculation time interval (described below) and simulation conditions, an optimal $\Delta x$ was found to be 500 m, which maximizes the numerical stability. In the given domain, 10 rows each at northern and southern boundaries are regarded as ocean. Hence, the effective terrestrial domain is composed of $200 \times 180$ cells. We found that the open waterbody (10 rows near boundaries) in front of the terrain on the course of airflow helps improve simulation results of the SB model and generate

smooth rain field over the terrestrial domain. Eastern and western boundaries are regarded as extended mountain ranges wherein the periodic boundary condition is applied. This setting induces the development of a mountain range parallel to the coastlines.

Whole landscape evolution modeling usually runs over the geologic time scale. Accordingly, calculation time interval is usually greater than a year. On the other hand, a storm event lasts at best a few days. To consider the time-variation of rainfall within each storm, the simulation should be implemented at least daily interval. Nevertheless, it is computationally exhausting

to implement the landscape evolution at daily time step over Ma time scales. To deal with this mismatching time scales, we adopted two concepts of (1) the dominant discharge and (2) dual time steps.

We first noted that not all storm events are geomorphologically effective. The streamflow corresponding to an event largely responsible for alluvial channel formation is referred to as the dominant discharge (Leopold et al., 1964). The dominant discharge is often considered similar to the effective discharge as well as the bankfull discharge, and corresponds to the

return period between 1 and 2 years (Knighton, 1998). In accordance with this notion, we only impose a single representative storm event (referred to as the dominant storm) within the return period $\Delta t_1$, while taking $\Delta t_1$ as the primary calculation time interval. The flow generated by the given storm is considered as the dominant discharge. We adopted $\Delta t_1 = 2$ years, which is close to the return period of dominant discharge reported in the field.





The dominant discharge is of a significant amount and so $\xi$. If such a large amount is imposed in a single time step, we may

encounter the over-erosion and the formation of depression zones, the area of which elevation $z$ lower than any neighboring cell. To prevent or minimize over-erosion, it would be necessary to divide the storm duration into multiple time steps. In this respect, we adopted another time step of $\Delta t_2$, smaller than $\Delta t_1$ as well as the duration of the dominant storm event, in discretizing equation (14). Accordingly, equation (14) is discretized as

$$\Delta z = u\Delta t_1 + \frac{1}{(1-\lambda)\rho_s} \sum_{n=1}^{D} (\xi_n - \xi_{in,n})\frac{\Delta t_2}{\Delta x} \tag{15}$$

where $\xi_n$ and $\xi_{in,n}$ are outgoing and incoming total sediment fluxes, respectively, at day $n$. Note that $\xi_n$ is calculated as the mass flux from the given cell toward the downstream cell determined by the improved GD8. Meanwhile, $\xi_{in,n}$ is the sum of fluxes originated from all neighboring cells (up to 8 cells) contributing flow toward the given cell. In this study, we adopted $\Delta t_2 = 1$ day. The dominant storm event may last for $D$ days, and equation (14) is solved for each day ($D$ iterations per $\Delta t_1$). By combining the concepts of the dominant discharge and dual time intervals, we can implement virtually daily simulations

while save computational resources substantially.

For the purpose of this study, an ideal simulation target can be a high and active mountain range (such as Andes and Himalaya) where orographic effect appears clearly. However, we considered the physical limit in the applicable peak altitude of the single-layer SB model. The peak altitude of the Olympic Mountains, USA, for which the SB model was evaluated in a previous study (Smith and Barstad, 2004), is about 2.4 km. Taking this as a reference, the tectonic uplift rate $u = 1$ mm/yr

is used for all simulations, which yields a peak elevation about 1.7 km (in most simulations) with a maximum of 2.7 km. In this way, we sought a reasonable compromise between the need to generate high mountain range and the model limitation. This uplift rate is imposed uniformly over the domain and constantly throughout the entire simulation duration of 5 Ma. The tectonic response to unloading patterns induced by spatial variations in precipitation is ignored. The other model parameter values are summarized in Table 1.

**Table 1.** Model parameters commonly used in all simulations

| Name | Symbol | Value |
|---|---|---|
| Porosity | $\lambda$ | 0.4 |
| Sediment particle density | $\rho_s$ | 2,650 kg m$^{-3}$ |
| Coefficient in equation (13) | C | 1,000 kg min$^{0.6}$m$^{-4.2}$ |
| Slope exponent in equation (13) | $\alpha$ | 1.7 |
| Discharge exponent in equation (13) | $\beta$ | 1.6 |
| Angle of repose | $\Theta$ | 32° |

Most landscape evolution modeling studies adopt random perturbations to reproduce dendritic stream network formation. Randomness can be imposed in various media such as the initial topography (e.g., Han et al., 2015) and surface material (e.g., Paik and Kumar, 2008). We adopted the former approach and the random noise, following a uniform distribution, is given in the





initial topography. The initial topography is nearly flat with a very mild slope imposed to make sure surface water flows toward the ocean. The given random perturbation is tiny enough relative to the given initial valley gradient, and so no depression

zone forms in the initial topography. We implemented simulations initiated with the common setting described above but with varying model complexities. Two classes of simulations, namely 'no feedback' and 'co-evolution' are presented below.

## 3.1 No-feedback simulations

No-feedback simulation results are obtained by running the LEGS model with spatially uniform rainfall, invariant over time. In no-feedback simulations, we focus on the effect of different rainfall excess rates $P$ on long-term topography evolution. In

the SB model, this can be seen as varying $P_o$ in equation (6) with zero wind velocity, which will negate the orographic effect (i.e., $P = P_o$). We applied three different scenarios of $P = 50, 100$, and 150 mm/day. We assumed these storms continue for 4 days ($D = 4$), hence the total rainfall depth for each event is 200, 400, and 600 mm, respectively. Each event is considered corresponding to a dominant storm, occurring in every two years ($\Delta t_1 = 2$ years). With this $\Delta t_1$, 2.5 million iterations are implemented for 5 Ma simulation time. Actual numerical simulations are implemented at daily resolution, i.e., 4 iterations per

every $\Delta t_1$. Hence, the computational load is 10 million iterations for 5 Ma simulation time for each scenario.

As the constant uplift is continuously imposed, the topographic relief increases over time in all cases, starting from zero (the sea level), and converges to a constant (Figure 1a). In this paper, the topographic relief or briefly relief refers to the gap between the peak elevation in the domain and the base level (which is zero as it refers to the sea level). As the greater $P$ is imposed, the topography converges to the lower relief. This agrees with results from an earlier experimental study (Bonnet and

Crave, 2003).

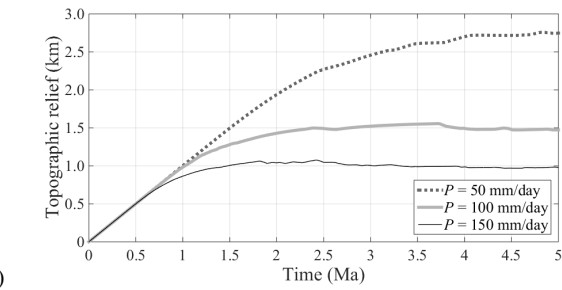

(a)

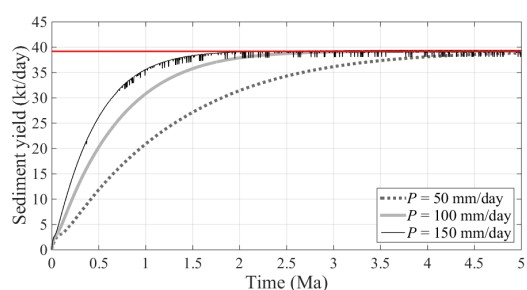

(b)

**Figure 1.** Results of no-feedback simulations. Time-variations of (a) the topographic relief and (b) the sediment yield from the entire domain are shown.

The sediment yield rapidly increases along with the topographic relief during the early stage of evolution (Figure 1b). Rapid stream network formation at the early stage (Paik and Kumar, 2008) accelerates the sediment yield. As simulations proceed, the sediment yield converges into the rate of tectonic mass increase by the uplift (shown as a red horizon in Figure 1b). This implies that the steady-state sediment yield is determined by the tectonic uplift rate, not rainfall. Although sediment yield

exhibits clear convergence (Figure 1b), the relief continuously varies (Figure 1a). This implies that our simulations produce





quasi-steady, instead of pure steady, landscapes. The strict definition of the steady state landscape requires no cell in the domain to show any elevation change over time. Referring to equation (14), this can be achieved if "the erosion rate is areally uniform and balances the rate of tectonic uplift" (Howard, 1994).

The balancing condition between the sediment yield and the uplift rate at the quasi-steady state helps explain distinct levels

of reliefs as the system approaches quasi-steady states (Figure 1a). To generate the same sediment yield at steady states, a high rainfall excess scenario is expected to pair with a low relief, as the sediment flux is given as the product of the runoff and gradient (equation (13)). This argument is also in accordance with an earlier study with the detachment-limited condition (Whipple and Tucker, 1999). Note that the inverse relationship between the imposed $P$ and the resultant relief is found nonlinear (as $P$ linearly grows from 50 to 100 and 150 mm/day, relief reduces nonlinearly from 2.7 to 1.5 and 1 km). This is due to the

nonlinear combination of runoff and gradient in equation (13).

One may notice that, in the scenario of the greatest rainfall ($P = 150$ mm/day), the sediment yield trend fluctuates over time. The episodic landslide events along coastlines can generate such fluctuations. Another mechanism causing the fluctuation is the formation and filling of depression zones. We found that the latter (depression zone) is mainly responsible for the fluctuations in our simulations. There is no depression zone initially, but they may appear during simulations as a result of over-erosion. Once

formed, it only receives sediment inputs, generates no sediment output, and so lowers the sediment yield from the domain. As a depression zone is filled with sediments, it disappears over time. In our simulation setting, the formation of depression zone is due to the over-erosion. To reduce over-erosion, it would be needed to adopt even a smaller $\Delta t_2$. This will further increase already large number of iterations or computational load. With $\Delta t_2 = 1$ day, the over-erosion is found only at the highest $P(=150$ mm/day) and in scenarios with lower $P$, no depression zones form and hence no fluctuations in sediment yield

graph. Orographic rainfall generated in our co-evolution simulations (to be shown below) is less than $P = 150$ mm/day as well. Hence, we adopt $\Delta t_2 = 1$ day throughout this study to save computational load.

### 3.2  Co-evolution simulations

We evaluated the long-term consequences of considering the orographic rainfall effect by coupling the LEGS and SB models. In co-evolution simulations, the rainfall field spontaneously evolves over time and space. We followed basic simulation settings

of no-feedback simulations: a dominant storm events lasting for a 4-day duration ($D = 4, \Delta t_2 = 1$ day), considered per every 2 years ($\Delta t_1 = 2$ years), and uniform tectonic uplift rate $u = 1$ mm/yr imposed continuously for the entire simulation time of 5Ma.

A number of simulation sets were designed with various atmospheric conditions. We evaluated the combined effects of the wind $\boldsymbol{v}$, the delay times ($t_c$ and $t_f$), and the temperature $T$ on the rain field generation and long-term topography. The wind

direction is considered consistently perpendicular to the mountain range, i.e., $U = 0$. As such, $\|\boldsymbol{v}\| = V$ and a wide range of $V$ was applied. $V$ is the vertically-integrated speed while the wind speed exhibits a profile often expressed as the logarithmic (Rossby, 1932) or power (Köhler, 1933) function of altitude. Therefore, $V$ value imposed in the SB model is considered greater than the ground wind speed. Delay times are dependent on $T$. Anders et al. (2008) reported that $t_d(= t_c + t_f)$ (in sec)





is inversely related with $T$ (in K), which can be approximated as

$$t_d = 58000 - 200T. \tag{16}$$

According to equation (16), three pairs of $t_d$ and $T$ are chosen for simulations: (1) $t_d = 600$ s, $T = 287$ K; (2) $t_d = 1200$ s, $T = 284$ K; and (3) $t_d = 1800$ s, $T = 281$ K. It has been known that the relative contribution of $t_c$ or $t_f$ in $t_d$ is insignificant for the generation of orographic rainfall (Smith and Evans, 2007). Therefore, we simply set $t_c = t_f (= t_d/2)$. For each pair, three levels of wind speed ($V = 8, 16, 24$ m/s) were tested, yielding nine scenarios in total. We imposed the base level rainfall $P_o = 20$ mm/day in equation (6). Each of these nine scenarios is considered as a dominant storm.

Finally, the environmental lapse rate $\gamma$ was carefully chosen to consider the physical limitation of the SB model. With $T$, $\gamma$ determines the moist layer height (equation (8)). Because no vertical variability is allowed in the SB model, applying the SB model to a topography higher than the moist layer is questionable. As described earlier, the maximum topographic height within which the SB model has been applied is 2.4 km. Considering this, we set $\gamma = -6.3$ K/km, so that the range of the moist layer depth among nine scenarios is between 2.32 and 2.44 km. The moist adiabatic lapse rate $\Gamma_m$ in equation (4) is given as $-6.5$ K/km in our theoretical simulations, referring to examples of Smith and Barstad (2004).

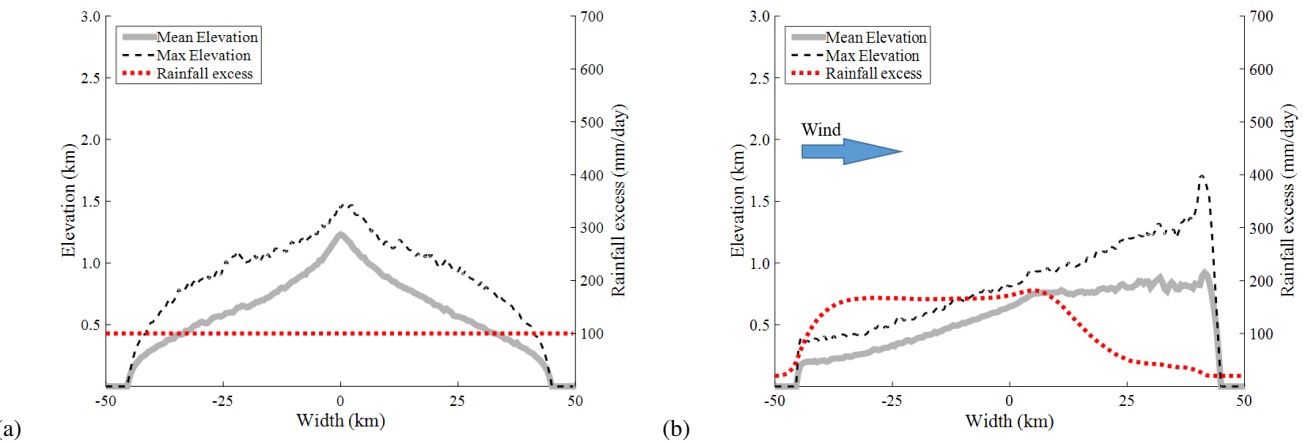

**Figure 2.** Transects of landscapes and rainfall fields after 5 Ma simulation times. Results from (a) no-feedback ($P = 100$ mm/day) and (b) co-evolution ($V = 16$ m/s, $t_d = 1200$ s, $t_* = 0.427$) simulations are shown.

Topography evolves in a dramatically different manner between no-feedback and co-evolution simulations, implying significant topography-climate feedbacks. For example, two simulations of 'no feedback' with $P = 100$ mm/day and 'co-evolution' with $V = 16$ m/s and $t_d = 1200$ s are compared in Figure 2. These are chosen in that rainfall depths accumulated over the total simulation period are similar. No-feedback simulations result in symmetric transects, whereas co-evolution simulations result in asymmetric transects. Their formation is directly related to the asymmetric rainfall distribution (Willett, 1999). Co-evolution simulations resulted in the peak being pushed toward the lee side. Such peak migration matches the postulation of Smith (2006), the physical experiments of Bonnet (2009), and earlier simulation studies with the detachment-limited condition





(Anders et al., 2008; Han et al., 2015). This will be further discussed in the next section. The SB model generates the rainfall

as a function of wind vector, local topographic gradient, delay time, temperature, and other meteorological variables. Unlike
the simpler assumption of rainfall as a monotonic function of elevation, the generated rain field exhibits gradual reduction as
the moist air penetrates deeper inland (Figure 3).

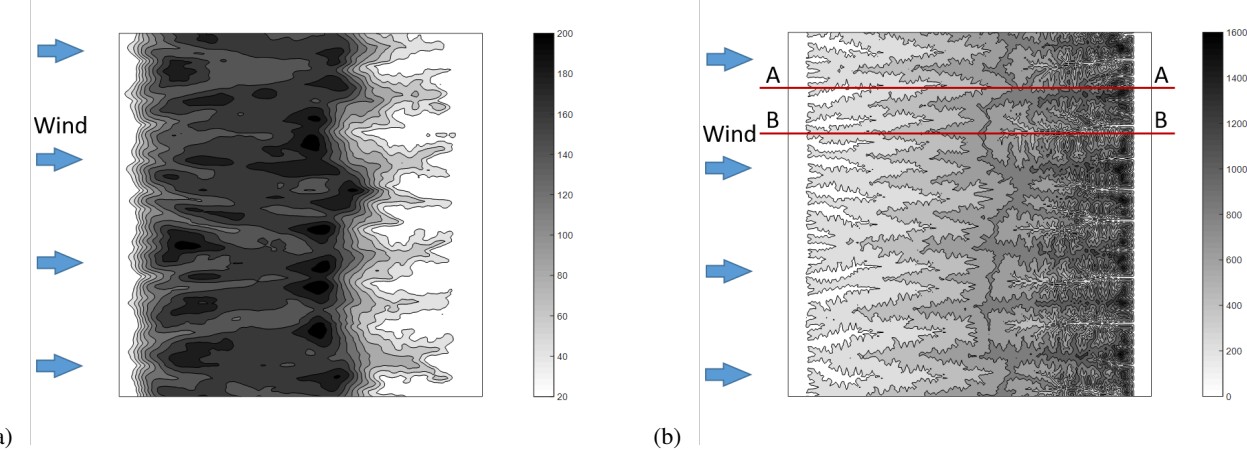

**Figure 3.** Contour map of simulated (a) rainfall field and (b) surface elevation from the co-evolution simulation with $V = 16$ m/s and
$t_d = 1200$ s after 5 Ma (its transect shown in Figure 2b). Numbers in the legend indicates the rainfall excess rate (mm/day) and surface
elevation (m), respectively.

## 4   Discussions

### 4.1   Asymmetry and relief in evolving topography

The formation of asymmetric topography is related with the time-evolution of rain field. On the initial flat surface, rainfall is
given uniformly as $P_o$ without any orographic effect. Channel incision initiates at coasts (outlets) and then migrates upstream.
This process should occur symmetrically across the initial transect. As the topography uplifts, the uniformity in the generated
rain field is broken and the rain field evolves in an asymmetric fashion (Figure 4a). Windward side receives more rainfall and
so more runoff as well as sediment yield than the lee side. As a result, the channels on the windward side propagate upstream

at a faster rate, compared to those on the lee side Figure 4b). This leads to the downward peak migration and a steeper slope
on the lee side (Figure 4c).

   As described above, the landscape experiences asymmetric erosion between the two sides (Figure 5). On the other hand,
the tectonic uplift is maintained spatially uniform. Topography at each time step is determined through the trade-off between
asymmetric erosion and uniform uplift. An interesting outcome of this complex adjustment is the development of the distinct

peaks, which can be shown as a high ridge-valley relief at the lee side (Figure 3b). These sharp peaks have been created by the
interference between different evolutionary processes on the windward and lee sides. Whereas the windward side experiences





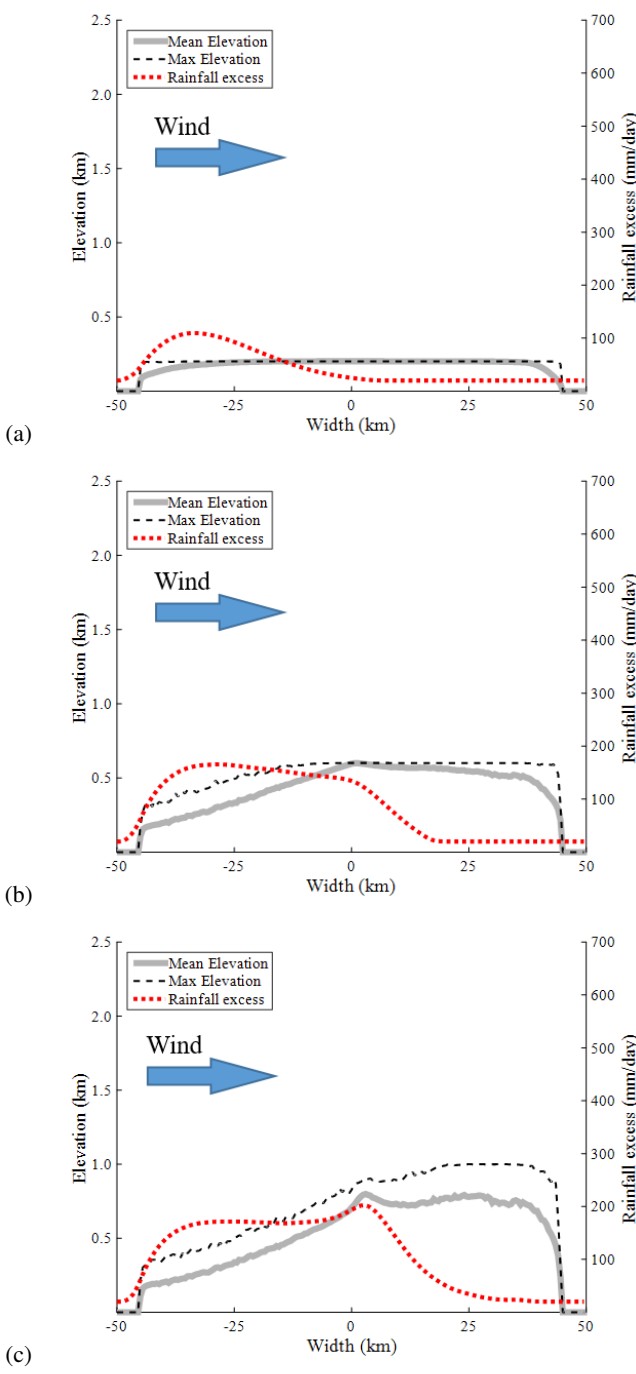

**Figure 4.** Transects of evolving landscapes and rainfall field from a co-evolution simulation after (a) 0.2, (b) 0.6, and (c) 1 Ma of simulation times. Simulation is implemented with $V = 16$ m/s and $t_d = 1200$ s. The results at 5 Ma are shown in Figure 2b and 3.



high erosion due to enhanced rainfall during the evolution, many peaks remain intact on the lee side due to much less rainfall while being elevated (Figure 4b). This results in a sharp contrast in elevation along the ridge line (Figure 3b). Therefore, such features are remainder of paleo-topography, similar to the monadnock.

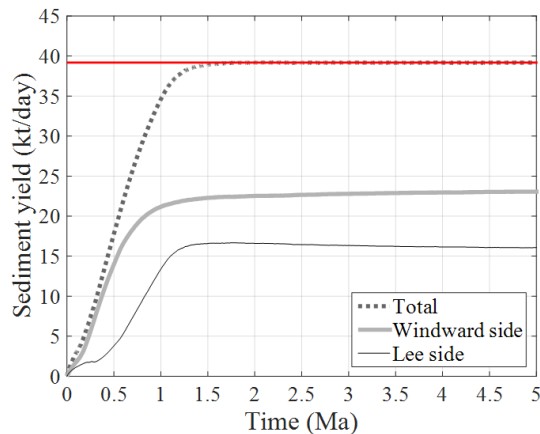

**Figure 5.** Time-variation of sediment yield from co-evolution simulation. Simulation is implemented with $V = 16$ m/s and $t_d = 1200$ s. Red horizontal line corresponds to the mass balance with the tectonic uplift rate.

Catchments on windward and lee sides develop alternately (Figure 3b). For a large catchment developed in the windward side, its channel head deeply penetrates beyond the domain center and toward the lee side (e.g., cross-section A-A in Figure 3b). The space between upper areas of two adjacent catchments from the windward side is taken as a niche where a major lee-side catchment develops. Along the valley line of a major catchment on the lee side (e.g., cross-section B-B in Figure 3b), the head of the lee side channel reaches close to the domain center. Through the topography-feedback on the local climate, 310   generated rainfall also exhibits alternate spatial pattern (Figure 3a).

    In their study for the detachment-limited condition, Roe et al. (2002) claimed that the orographic rainfall reduces relief. In their model, rainfall increases monotonically with elevation and reaches a maximum at the peak elevation, which leads to a large peak incision and hence lower relief. However, the real rainfall pattern induced by the orographic effect is often far from such monotonic increase with elevation (e.g., Bookhagen and Burbank, 2006). The SB model generates the rainfall distribution 315   much more complex and diverse, far from monotonic gradient. As a result, our simulations generate relief comparable to, or even greater (higher peak elevation) than, the relief obtained from no-feedback simulations for similar accumulated rainfall depths (e.g., Figure 2). Depending on local atmospheric and topographic conditions, rainfall concentrates on different altitudes, influencing relief in different ways.

### 4.2   Roles of wind speed and total delay time in co-evolution

In the co-evolution modeling, the domain-averaged rainfall amount increases over time from the initial rate of $P_o$ (Figures 6a, b). Such increase is in accordance with the relief growth and consequent feedback on rainfall. As shown in equation (3), the




wind speed together with the topographic gradient is the primary factor that controls the rainfall. Essentially all rainfall depth generated beyond $P_o$ is due to the orographic effect (equation (6)), which becomes activated only if a moist air parcel is pushed by wind and climbs uphill. Clearly, orographic rainfall shall be nil when $V = 0$, and the generated rainfall amount tends to

increase as a greater wind speed is imposed (Figure 6a).

For a greater wind speed $V$, rainfall increases at a faster rate and converges to a greater amount. Such relationship between $V$ and $P$ is, though, found nonlinear. Physical upper limit appears to exist as $V$ greater than a certain value contributes no further increase in $P$. With $t_d = 1200$ s and $T = 284$ K, $V = 16$ m/s seems to be the limit (Figure 6a) although a greater rainfall is possible with a smaller $t_d$ and warmer weather. Because the generation of $P$ depends on both $V$ and the topographic gradient

(equation (3)), to understand this nonlinear relationship and the limit in $P$, it is needed to understand the long-term effect of $V$ on topography through rainfall, and the topography feedback on $P$, which is discussed below.

The resulting relief tends to be lower with a greater $V$ (Figure 6c). A similar tendency can be claimed for the mean elevation (Figure 6e). These can be explained with the same reasoning described for no-feedback simulations (Figure 1), i.e., greater wind speed is associated with more rainfall, and so it must be paired with a lower gradient to yield a sediment flux matching

the mass generated at the imposed uplift rate. For a small $V$ (e.g., 8 m/s), a small $P$ is generated, which results in a high relief (Figure 6c). Recall that the high relief can give a positive feedback on generating $P$. As $V$ increases, the relief tends to decrease, which gives a negative feedback on $P$. So, as $V$ grows, there is a trade-off between the direct positive feedback on $P$ and indirect (via a smaller relief) negative feedback on $P$. In two cases of $V = 16$ and 24 m/s shown in Figure 6a, these effects have coincidently become identical, which results in the same $P$.

The inverse relationship between $P$ and the relief is described above. Interestingly, those two runs with $V = 16$ and 24 m/s, while producing similar rainfall, show relief difference (Figure 6c). This can be explained by another mechanism: a greater wind speed carries rainfall further uphill, toward the lee side (compare Figures 2b and 7a). This results in a greater erosion over the peak, which also contributes to lower relief.

Opposite to the $V$ vs. $P$ relationship, $P$ decreases as $t_d$ grows (Figure 6b). This is because a part of hydrometeors generated

within the domain falls out of the domain in proportion with $t_d$. For example, the scenario with $V = 24$ m/s and $t_d = 1800$ s embeds 43 km horizontal displacement between locations of hydrometeor formation and rainfall. Because a shorter delay time is related with a greater rainfall, one may also expect a robust relationship between $t_d$ and the steady-state relief. Indeed, the robust relationship was found between $t_d$ and the mean elevation (Figure 6f). However, similar relationship was not consistently obtained between $t_d$ and relief (Figure 6d). Although we find the inverse relationship between $t_d$ and the rainfall amount

(domain averaged) (Figure 6b), a high rainfall induced by a short $t_d$ concentrates on downhill (compare Figures 2b and 7b), which gives little effect on the peak elevation. Hence, no general relationship between $t_d$ and relief can be established.

As discussed, both wind and $t_d$ play primary roles in rain field generation. In fact, the combined effect of $\|\boldsymbol{v}\|$ and $t_d$, rather than their individual effects, controls the co-evolutionary system. In this regard, the non-dimensional delay time $t_*$ (equation (1)) was claimed as a representative measure in shaping the resultant topography (Anders et al., 2008). However,

our co-evolution modeling shows that steady-state topography can substantially differ even for an identical $t_*$. For example, Figure 7 illustrates two simulations with the same $t_*$ of 0.213 but very different resultant topography. The dramatic difference





**Figure 6.** Variations of (a-b) domain-averaged rainfall excess, (c-d) topographic relief, and (e-f) mean elevation during co-evolution simulations with different (a, c, e) $V$ ($t_d$ fixed as 1200 s) or (b, d, f) $t_d$ ($V$ fixed as 16 m/s).





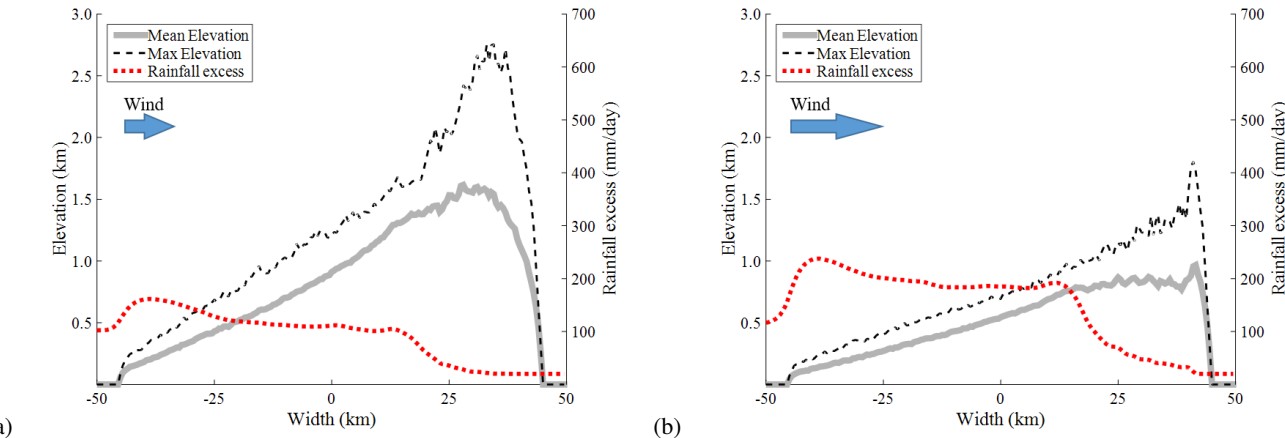

**Figure 7.** Transects of landscapes and rainfall fields from co-evolution simulations after 5 Ma simulation times. Simulation settings are: (a) $V = 8$ m/s, $t_d = 1200$ s and (b) $V = 16$ m/s, $t_d = 600$ s.

is indebted to multiple effects of $V$ and $t_d$. While they give similar effect on the horizontal displacement of rainfall, they act oppositely in producing overall rainfall amount: increased $V$ induces greater rainfall (domain average) whereas increased $t_d$ reduces rainfall (Figure 6).

As described previously, the pure steady state can be claimed if every cell in the domain exhibits no elevation change over time. This would be referred to as the dynamic equilibrium. In macro-perspective, the dynamic equilibrium can be inferred when the relief becomes steady (Hack, 1960; Willgoose, 1994). In co-evolution simulations, generated rainfall amount converges to a constant, which was claimed as another indicator of a steady state (Han et al., 2015). In our simulations, the rainfall (Figures 6a, b), relief (Figures 6c, d), mean elevation (Figures 6e, f), and sediment yield (Figure 5) all converge to constants.

However, the convergence times are all different among these criteria, implying that the pure steady states have not been achieved.

### 4.3 Profile concavity

Along a natural stream, the local slope $\|\nabla z\|$ and drainage area $A$ are associated as (Flint, 1974)

$$\|\nabla z\| \propto A^{-\theta} \tag{17}$$

where $\theta$ is the concavity index. For alluvial channels, $\theta$ is mostly found positive (Howard, 1980; Tucker and Whipple, 2002; Byun and Paik, 2017), which indicates a concave profile. No-feedback simulations have well reproduced concave profiles (e.g., Willgoose et al., 1991b; Willgoose, 1994; Paik, 2012). Here, a basic question is raised about the role of orographic rainfall in the profile formation. Roe et al. (2002) reported the impact of orographic rainfall on channel concavity in the 1D context. Below, we discuss the formation of channel profiles in a more general 3D landscape context.





Channel profiles at quasi-steady states, simulated from our co-evolutionary model, are mostly concave but we found that the profile concavity on the windward side becomes lower as $V$ or $t_d$ increases (Figure 8). We can explain such variation with the scaling relationship below (Willgoose et al., 1991b)

$$\theta = \alpha^{-1}[\phi\beta - 1]. \tag{18}$$

In equation (18), $\phi$ is the exponent of the power-law relationship between the streamflow $Q$ and drainage area $A$, i.e., $Q \propto A^{\phi}$.
A more comprehensive equation, considering the hydraulic geometry (Leopold and Maddock, 1953), Hack's law (Hack, 1957), and downstream fining (Brush, 1961) relationships, was derived by Byun and Paik (2017). Here, we use a briefer equation (18) for a simpler explanation. With fixed sediment transport parameters $\alpha$ and $\beta$ (used in equation (13)), equation (18) implies that the concavity $\theta$ is proportional to $\phi$. Considering that the flow within a catchment propagates through a stream network downstream, if rainfall concentrates downstream high $\phi$ is expected (significant downstream increase of streamflow). As $V$ or
$t_d$ increases, the rainfall distribution moves upstream which will result in a lower $\phi$. From equation (18), hence, a greater $V$ or $t_d$ can lead to a decreased concavity. If $\phi$ is excessively low (and hence $\phi\beta < 1$) even a convex profile may form.

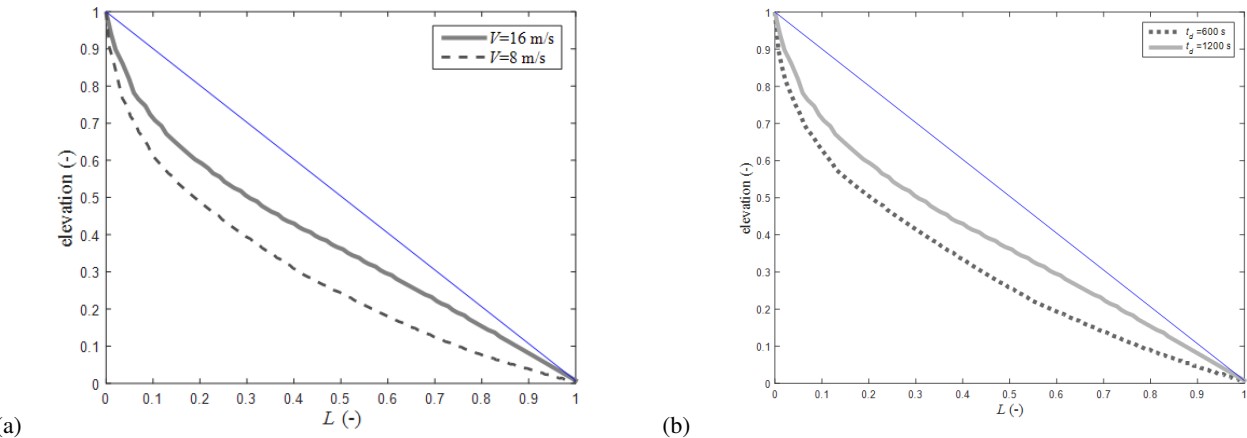

**Figure 8.** Channel profiles after 5 Ma from simulations with (a) different $V (t_d = 1200$ s) and (b) different $t_d (V = 16$ m/s). The profile of the main channel of the largest catchment in each simulations is shown. For comparison, profiles are normalized with maximum elevation and lengths.

### 4.4   Limitations

It is important to note the limitations of the SB model in reproducing complex rainfall patterns in mountainous terrain. A critical weakness, relevant to our modeling approach, is that the SB model considers only single values of $T$ and $V$ (i.e.,
spatially uniform) for the entire domain. Ideally, a 2D gridded or even 3D weather model can be coupled with the surface process model. But running such a model for a geologic time scale currently faces a computational challenge (we ran 10 million iterations for each scenario).





We have demonstrated that generated rainfall decreases as the delay time grows (Figure 6b). This accounts for missing hydrometeors generated inside but fell out of the domain. But, there are also hydrometeors formed out of the domain and blown into the domain. Such amount yet to be counted for more realistic simulations. For counting such effect, it needs to be deeply investigated to what spatial extent such effect shall be considered.

Finally, we face a basic question whether simulation results are testable. There is absolute lack of documented co-evolution record. While this is a fundamental issue in surface process modeling community, we need cooperative efforts to address it.

## 5  Conclusions

As the close linkage between topography and climate has been acknowledged, numerical modeling of their co-evolution has become an important subject. In this study, a sophisticated numerical model is developed for the given purpose, and results from 'no-feedback' and 'co-evolution' simulations are compared. Detailed simulation results provide insights on the feedbacks and controls in co-evolution. Major scientific contributions from our theoretical study can be summarized below.

– Evolving topography carries climatic and geomorphic footprints as shown in asymmetric transects and high valley relief on the lee side. Orographic rainfall affects resulting topographic relief in complex ways as spatial rainfall distribution greatly varies with meteorological conditions.

– Concavity in channel profile on the windward side becomes lower as the wind speed or delay time increases. This is explained with the known scaling relationship between the stream flow and upslope area, which is associated with the spatial rainfall distribution, driven by wind and topographic gradients.

– Evolved topography under the same non-dimensional delay time (equation (1)) can significantly differ because of different roles of the wind speed and total delay time.

In most landscape evolution modeling studies, rainfall has been considered as an input. On the basis of results shown in the present study, we suggest that the modeling community shall reconsider this practice: we should incorporate local climate dynamics in landscape evolution. Probably, this is the time to revise the terminology of the whole landscape evolution modeling as modeling the evolution of the topography-climate coupled system. It is desired to further explore the roles of model (particularly meteorological) parameters and their combinations. In this sense, simulation results presented here are examples calling for the future research needs in co-evolution modeling.

*Code availability.*  The coupled simulation model LEGS used for the examples given in this paper will be openly available for non-commercial purposes from our website (http://river.korea.ac.kr).





**Appendix A**

The sediment transport equation (13) adopted in this study follows the classical formula, widely applied over a century. The first literature on this functional form traces back to du Boys (1879), whose sediment transport equation follows

$$q_s \propto \tau_b(\tau_b - \tau_c) \tag{A1}$$

where $\tau_b$ is the bed shear stress and $\tau_c$ is to encounter the initiation of motion (similar to but not exactly the same as the critical shear stress).

The bed shear stress can be estimated as $\tau_b = \rho g H \|\nabla z\|$ where $\rho$ is the density of fluid and $H$ is the flow depth. In analogy, we can suggest $\tau_c = \rho g H_c \|\nabla z\|$ where $H_c$ is the threshold depth corresponding to the initiation of the motion for the given slope $\|\nabla z\|$. Substituting these in equation (A1) gives

$$q_s \propto \|\nabla z\|^2 (H^2 - H_c H). \tag{A2}$$

If we take the Chezy formula, $H \propto \|\nabla z\|^{-1/3} q^{2/3}$. Similarly, $H_c \propto \|\nabla z\|^{-1/3} q_c^{2/3}$ where $q_c$ is the threshold value of $q$ corresponding to $H_c$ and $\tau_c$. Then, equation (A2) can be rewritten as

$$q_s \propto \|\nabla z\|^{1.33} (q^{1.33} - q^{0.67} q_c^{0.67}). \tag{A3}$$

Instead of the Chezy, the Manning formula can also be used, i.e., $H \propto \|\nabla z\|^{-0.3} q^{0.6}$ and $H_c \propto \|\nabla z\|^{-0.3} q_c^{0.6}$. Substituting these in equation (A2) gives

$$q_s \propto \|\nabla z\|^{1.4} (q^{1.2} - q^{0.6} q_c^{0.6}). \tag{A4}$$

Equation (A2) states $q_s$ as a function of $H$. By transforming equation (A2) into equation (A3) or (A4), $q_s$ is expressed as a function of $q$. This is practically advantageous in that the evaluation of flow discharge is easier than the flow depth, particularly in the context of landscape evolution modeling.

Austrian engineer Schoklitsch implemented laboratory experiments to evaluate this type of equations with the grain size between 1 and 2 mm. The equation of Schoklitsch published in 1934 (introduced in English by Shulits (1935)) is

$$q_s = C_o d^{-0.5} \|\nabla z\|^{1.5} (q - q_o) \tag{A5}$$

where $q_o$ is the threshold value of $q$ for the initiation of motion and $d$ is the grain size. The coefficient $C_o$ has a complex dimension. In English units ($q_s$ in lb/sec/ft, $d$ in inches, $q$ in ft$^3$/sec/ft), $C_o = 86.7$lb in$^{0.5}$ft$^{-3}$ was suggested from flume experiments (see Shulits (1935)), which is about 7000 kg mm$^{0.5}$m$^{-3}$ in metric units ($q_s$ in kg/sec/m, $d$ in mm, $q$ in m$^3$/s/m).

Above equations (A3), (A4), and (A5), if the threshold term is neglected, commonly express the sediment transport rate for non-cohesive bed streams as the function of local slope and flow discharge, suggesting the form of equation (13). Quantifying the initiation of motion is difficult, especially for small grains such as sands (Byun and Paik, 2017). Movement of sand particles has been observed even at very weak flow in flume experiments (e.g., Shvidchenko and Pender, 2000). Considering such a large



uncertainty associated with it, the threshold term is often neglected when applying these equations to the scale of landscape
evolution (Willgoose et al., 1991a; Tucker and Slingerland, 1994; Dietrich et al., 2013).

Other sediment transport equations such as Shields, Einstein-Brown, and Kalinske can also be expressed in the form of
equation (13) (Raudkivi, 1967). Taking the Chezy or Manning formula, the Shields equation can be expressed as

$$q_s \propto \|\nabla z\|^{1.67} q^{1.67} \text{(Using Chezy)} \qquad \text{or} \qquad q_s \propto \|\nabla z\|^{1.7} q^{1.6} \text{(Using Manning)}. \qquad \text{(A6)}$$

Derivation from Einstein-Brown equation is given in Henderson (1966) and later by Willgoose et al. (1991a) in a more detail
as

$$q_s \propto \|\nabla z\|^2 q^2 \text{(Using Chezy)} \qquad \text{or} \qquad q_s \propto \|\nabla z\|^{2.1} q^{1.8} \text{(Using Manning)}. \qquad \text{(A7)}$$

As shown in this appendix, equation (13) can be derived from many existing sediment transport formulae. It has also been
directly supported by flume experiments. While all these approaches result in the form of equation (13), they differ in the
values of exponents $\alpha$ and $\beta$. For example, on the basis of equation (A3), $\alpha = \beta = 1.33$ while equation (A4) implies $\alpha = 1.4$
and $\beta = 1.2$. Even greater values are shown in equation (A6) or (A7). Another experimental study by MacDougall (1933) is
also noteworthy where the same functional form of equation (13) was suggested with a range of $\alpha$ between 1.25 and 2 while
$\beta$ is kept as unity. From literature, we can see that the range of $\alpha$ and $\beta$ are $1 < \alpha < 2$ and $1 < \beta < 2$. Details of these ranges
have been studied by Prosser and Rustomji (2000). Comparing equations (13) and (A5), the coefficient $C$ in equation (13) is
interpreted as $C = C_o d^{-0.5}$ given $\alpha = 1.5$ and $\beta = 1$. As $\alpha$ and $\beta$ vary, the value and dimension of $C$ in equation (13) are
subject to vary.

*Author contributions.*  KP acquired funding, designed the simulation, analyzed results, did scientific interpretations, and wrote the manuscripts.
KP and WK developed the model code. Through his Master thesis, WK proposed to couple the SB and LEGS models, and tested the early
ideas about co-evolution.

*Competing interests.*  The authors declare that they have no conflict of interest.

*Acknowledgements.*  This work was supported by the National Research Foundation of Korea (NRF) grant funded by the Korea government
(MSIT) (No. 2018R1A2B2005772).



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
