# Peer review of "Simulating the evolution of the topography-climate coupled system"

_Hydrology and Earth System Sciences, 2020_

## Referee Comment (RC1) · Arnaud Temme (Referee) · 19 Nov 2020

The manuscript by Paik and Kim presents a novel model that better resolves the feedback between rainfall and topography than has been done before, and extracts important lessons from simulations with that model. Specifically, Paik and Kim provide a formulation of rainfall patterns that is more responsive to topography than existing formulations, which allows a close and realistic coupling between the development of a mountain range under the influence of uplift and erosion on the one hand, and the rainfall that drives the erosion on the other hand. This close coupling allows the authors to ask how different aspects of the rainfall-causing atmosphere affect orogenesis, and they arrive at the conclusion (among others) that overall wind speed arriving at the mountain front has a different impact on the rainfall field and hence on topographic

development than the time needed to develop rain out of a rain-prone airmass. Substantial impacts on stream profile concavity are also explored. I feel this is an important step beyond what we knew before, and allows us to ask new questions that we could not ask before.

The manuscript is very well written and presented, and illustrated with figures and tables that are all useful. Particularly, the pace of explaining new model components is very measured and I could follow along as a result of this paced introduction. However, I would like to note that the atmospheric component of the combined model is so new to me that I recommend relying on comments of additional reviewers to assess that component's validity.

I feel that the manuscript is very near a level that will allow publication. However, I do have a few small suggestions. Most of these are aimed at making a geomorphologist-reader have an easier time understanding the novel atmospheric component. I present them here in the order found in the manuscript.

l18 first mention of hydrometeors - perhaps explain this to the uninitiated (is it raindrops, or raindrops and/or snowflakes and/or hail, for instance).

l40 rainfall depth as well is an unusual term (to me at least). Please add a few words to explain.

l69 "wonder" - do you perhaps mean "question" ?

l92 non-orographic large-scale vertically integrated condensation rate - so, does that mean through convectional uplift? I am not sure, but if it is, maybe that is worth mentioning to clarify the contrast with orographic uplift.

l136 "the entire landscape has been assumed..." - not quite. Models by Schoorl and by Lague, for instance, deal with both conditions, as far as I understand. I placed references below.

l168 how do you deal with sediment eroding off the mountain range and entering the
sea-cells? Is it assumed to disappear? The remainder of your text does suggest that, but it would be good to mention that here.

l184 "is of a significant amount and so zdeta" . I suggest: "and so IS zdeta"

l207 Please specify the amount of noise (perhaps by providing the ranges of the uniform distribution), and I also suggest to confirm in the text that the noise is not spatially auto-correlated.

l208 Please specify the initial slope value. This is especially relevant since earlier work has shown extreme sensitivity to the initial slope (in non-uplifting settings at least). I think that work used the Child model and the original may be in a paper first-authored by G.E. Tucker, but I couldn't find it anymore for you.

l211 At this point, you have explained the model in mathematical detail. As noted before, I feel that that is well done. However, I would really appreciate it if you could provide a figure about here that shows how the rainfall works out in model simulations. Perhaps showing the amount and net direction of descent of precip from the airmass moving over a hypothetical topography would work best, but any output that shows a bit of what is calculated before getting to the excess rainfall that you show from here on out, would be welcome.

l242 "landslide events" I don't believe you have yet told us that you simulate landsliding - or how it is done. Are you simply using a threshold slope gradient?

l243 "fluctuations" . That process is also discussed in Temme, 2006. I provide the reference below, feel free to disregard.

l262-264 I felt that this information is perhaps better placed in the model description section, and is distracting here.

Figure 3: The legend font is too small here. I noticed that the font size varies substantially among figures. Can you choose one font size and apply it to all figures?

Figure 3 caption: Numbers in the legend indicate (no "s")

l305-310 Can you here compare with overall topographic features in some of the world's mountain ranges that are closest to your condition that U=0? I believe the Andes have U=0, approximately. Do the Andes (or other mountains) match your simulated topography with feedbacks better than they do the topography without feedbacks, or even better than the topography with feedbacks from the Roe at al model? If this is not possible, you might consider here pointing at the difficulty of making comparisons with existing orogens.

l327 I suggest " was found to be nonlinear"

l344 Opposite: I sugggest "In contrast"

l345/6 Visualizing this rainfall displacement through wind and delayed formation is exactly what would work really well in a "Figure 0" as mentioned above.

l357 " indebted" : i suggest "due"

l398 I agree with your sentiment that cooperative efforts are needed to document co-evolution. If possible, can you share any thoughts what that could look like? I can imagine that thermochronology, for instance, can help constrain the topographic side of the co-evolution. Is the problem mostly with reconstructing spatiotemporal patterns of rainfall?

References: Davy, P., & Lague, D. (2009) Fluvial erosion/transport equation of landscape evolution models revisited. Journal of Geophysical Research B: Solid Earth, 114(3). Retrieved from http://www.scopus.com/inward/record.url?eid=2-s2.0-75849161206&partnerID=40&md5=f525702d700153179701b74d747455e0 Schoorl, J. M., Sonneveld, M. P. W., & Veldkamp, A. (2000). Three-dimensional landscape process modelling: The effect of DEM resolution. Earth Surface Processes and Landforms, 25(9), 1025–1034. Temme, A. J. A. M., Schoorl, J. M., & Veldkamp, A. (2006). Algorithm for dealing with depressions in dynamic landscape evolution models. Computers and Geosciences, 32(4). https://doi.org/10.1016/j.cageo.2005.08.001

<please note my choice to reveal my identity. I reveal my identity when I mention work that I was involved in myself, so that my self-suggesting is visible to the community and can be criticized if required, and so that I can tell authors that I in no way insist on my work being cited if they feel that it is not valuable to their manuscript>

---

## Referee Comment (RC2) · Anonymous Referee #2 · 29 Dec 2020

[referee-annotated manuscript omitted]

---

## Referee Comment (RC3) · Omer Yetemen (Referee) · 12 Jan 2021

This paper is very interesting for the landscape evolution community. It is well written.

In my opinion, this is an interesting and valuable contribution to geomorphology. I have minor comments for the paper.

L172-173. "Whole landscape evolution modeling usually runs over the geologic time scale. Accordingly, calculation time interval is usually greater than a year. On the other hand, a storm event lasts at best a few days."

Landscape evolution models not necessarily work on annual time scale. Currently, there are modelling examples run on hourly precipitation and daily precipitation intervals (e.g., Yetemen et al. 2015).

[Figure]

L196-197. "For the purpose of this study, an ideal simulation target can be a high and active mountain range (such as Andes and Himalaya) where orographic effect appears clearly."

Orographically enhanced precipitation can be more obvious in these mountain range; however, it can be effective in intracontinental semiarid locations too (Wainwright, 2005).

Wainwright, J. (2005). Climate and climatological variations in the Jornada experimental range and neighbouring areas of the US southwest. Advanced Environmental Monitoring and Modelling, 2, 39–110.

L208-210. "The initial topography is nearly flat with a very mild slope imposed to make sure surface water flows toward the ocean. The given random perturbation is tiny enough relative to the given initial valley gradient, and so no depression zone forms in the initial topography."

A bit more clarification is needed!

What is the direction of flow, D8-steepest descent? So, the perturbation must ensure the some of the side slopes of the cells must be steeper than those of the profile slope. Otherwise, all flow drains straight downward, and you cannot generate meandering or dendritic channels!

L213. "No-feedback simulation results are obtained by running the LEGS model with spatially uniform rainfall, invariant over time."

Do the authors mean that a single precipitation pulse is repeating during the entire simulation?

L277. For Figure 2b, can the authors provide/add 3D (oblique) view of the landscape? I am very curious to see this simulated topography. Following figure, 3b, 2D figure shows the densification in the counter lines but too black and height is not clear!

The mean elevation profile of Figure 2b is extremely asymmetric! Huge cliff on the leeward side, just about 5 km length. I am very surprised to see such asymmetric behavior when I compare with previous papers (Goren et al. 2014; Han et al., 2014; Han et al., 2015). Is it artifact of the model?

Goren et al. 2014. Coupled numerical–analytical approach to landscape evolution modeling. Earth Surf. Process. Landforms 39, 522–545.

L296-7. "This leads to the downward peak migration and a steeper slope on the lee side (Figure 4c)".

This definition can be valid for bare soil (as seen in lab experiments of Bonnet and Crave, 2003); however, in vegetated environment this may not be correct. Enhanced vegetation on the windward side as a result of more rainfall may impede erosion. The steepest side is defined by the competition between the erosion inhibition by vegetation and enhanced erosive power of increasing rainfall!

---

## Author Comment (AC1) · 3 Feb 2021

We thank Arnaud Temme (referred to as AT in the following) for thoughtful review comments. In particular, we are very pleased to find that AT recognizes the value of our study. We also appreciate line-by-line comments provided, which are very helpful to further improve the quality of this paper. Our reply to each point is listed below. Any change we mention below will be reflected in the revised manuscript which is allowed to be submitted by HESS in the next step.

**AT: l18 first mention of hydrometeors - perhaps explain this to the uninitiated (is it raindrops, or raindrops and/or snowflakes and/or hail, for instance).**

Reply: According to the American Meteorological Society, the 'hydrometeor' is defined

as "any product of condensation or deposition of atmospheric water vapor, whether formed in the free atmosphere or at the earth's surface; also, any water particle blown by the wind from the earth's surface." We have considered writing this definition in the text but decided not to do for two reasons: (1) this is a widely known terminology and (2) writing the definition requires a separate sentence, which can pause the smooth flow of the logic of the given paragraph.

**AT: l40 rainfall depth as well is an unusual term (to me at least). Please add a few words to explain.**

Reply: We will replace it with 'rainfall amount.'

**AT: l69 "wonder" - do you perhaps mean "question" ?**

Reply: Yes, we will correct it.

**AT: l92 non-orographic large-scale vertically integrated condensation rate - so, does that mean through convectional uplift? I am not sure, but if it is, maybe that is worth mentioning to clarify the contrast with orographic uplift.**

Reply: For clarification, we will revise the sentence as "Here, $S_o$ is the background condensation rate, driven by large-scale (synoptic) vertical wind component. If the orographic effect is nil (either $\vec{v}$ or $\nabla z$ is zero), $S = S_o$."

**AT: l136 "the entire landscape has been assumed..." - not quite. Models by Schoorl and by Lague, for instance, deal with both conditions, as far as I understand. I placed references below.**

Reply: Thanks for the reference. We agree that the statement cannot be applicable for 'all' modeling studies. We will revise the sentence as "In most theoretical modeling studies, however, the entire landscape has been assumed as one of these conditions."

**AT: l168 how do you deal with sediment eroding off the mountain range and entering the sea-cells? Is it assumed to disappear? The remainder of your text**

**does suggest that, but it would be good to mention that here.**

Reply: We will add a following sentence: "Sediments eroded over the landscape, entering the surrounding ocean, are considered permanently lost, i.e., sea floor is assumed as of indefinite depth."

**AT: l184 "is of a significant amount and so zdeta" . I suggest: "and so IS zdeta"**

Reply: Will be fixed

**AT: l207 Please specify the amount of noise (perhaps by providing the ranges of the uniform distribution), and I also suggest to confirm in the text that the noise is not spatially auto-correlated.**

Reply: As suggested, it will be revised as "We adopted the former approach and the random noise, following a uniform distribution (not spatially auto-correlated), is given in the initial topography. The initial topography is nearly flat with a very mild slope ($10^{-5}$) imposed to make sure surface water flows toward the ocean. The given random perturbation ($\pm 5 \times 10^{-4}$ m) is tiny enough relative to the given initial valley gradient, and so no depression zone forms in the initial topography."

**AT: l208 Please specify the initial slope value. This is especially relevant since earlier work has shown extreme sensitivity to the initial slope (in non-uplifting settings at least). I think that work used the Child model and the original may be in a paper first-authored by G.E. Tucker, but I couldn't find it anymore for you.**

Reply: See above reply.

**AT: l211 At this point, you have explained the model in mathematical detail. As noted before, I feel that that is well done. However, I would really appreciate it if you could provide a figure about here that shows how the rainfall works out in model simulations. Perhaps showing the amount and net direction of descent of precip from the airmass moving over a hypothetical topography would work best, but any output that shows a bit of what is calculated before getting to the**

**excess rainfall that you show from here on out, would be welcome.**

Reply: Upon the request, we generate a new figure as attached. This figure will be added in the revised manuscript. Figure caption would be "Schematic showing orographic rainfall over the 3-d bird view of topography. Results after 5 Ma from co-evolution simulations with $V$=16 m/s and $t_d$= 1200 s (shown in Figure 2b). Horizontal displacement of raindrop from generation to falling location is given as $t_d \times V$."

**AT: l242 "landslide events" I don't believe you have yet told us that you simulate landsliding - or how it is done. Are you simply using a threshold slope gradient?**

Reply: Yes, it is stated in L149-152 as "Shallow landslide is considered in the simple way, previously applied by Tucker and Slingerland (1994); any slopes greater than the angle of repose $\Theta$ shall fail, translating upslope mass at the rate $q_f$ to downhill until the local slope reaches $\Theta$. This contribution is combined with the fluvial sediment transport to form the total sediment flux $\xi(= q_s + q_f)$ at every cell."

**AT: l243 "fluctuations" . That process is also discussed in Temme, 2006. I provide the reference below, feel free to disregard.**

Reply: Yes, the variation of SDR in Temme (2006) is similar to what we see in this study. The reference will be cited.

**AT: l262-264 I felt that this information is perhaps better placed in the model description section, and is distracting here.**

Reply: We concur and will relocate this sentence to section 2.

**AT: Figure 3: The legend font is too small here. I noticed that the font size varies substantially among figures. Can you choose one font size and apply it to all figures?**

Reply: We will reproduce all figures with a consistent font size.

**AT: Figure 3 caption: Numbers in the legend indicate (no "s")**

Reply: Will be fixed.

**AT: l305-310 Can you here compare with overall topographic features in some of the world's mountain ranges that are closest to your condition that U=0? I believe the Andes have U=0, approximately. Do the Andes (or other mountains) match your simulated topography with feedbacks better than they do the topography without feedbacks, or even better than the topography with feedbacks from the Roe at al model? If this is not possible, you might consider here pointing at the difficulty of making comparisons with existing orogens.**

Reply: We appreciate this insightful comment. The Andes could be a great target to be examined. We, however, determine that such comparison with a real topography at this stage would be premature in a publishable work, although this could be a great topic for future study without a doubt. There are still technical improvements needed from the numerical model side as we stated in the limitation section. We also added the role of vegetation in that section as "The scope of the present study is the bare soil landscape. In reality, vegetation has played a significant role in shaping the Earth's terrestrial surface. Incorporating vegetation dynamics could make co-evolution results much more complicated, as their feedback mechanisms have complex links with water, solar radiation, nutrient, carbon, sediment (soil), topography, etc. Modeling vegetation dynamics in the landscape framework has been actively studied in the last decade (e.g., McGrath et al., 2012; Yetemen et al., 2015). With ongoing efforts we will be eventually able to incorporate vegetation dynamics into the present co-evolution framework."

**AT: l327 I suggest " was found to be nonlinear"**

Reply: Will be fixed.

**AT: l344 Opposite: I sugggest "In contrast"**

Reply: Will be fixed.

**AT: l345/6 Visualizing this rainfall displacement through wind and delayed forma-**

**tion is exactly what would work really well in a "Figure 0" as mentioned above.**

Reply: See above reply. A new figure (attached) would satisfy this comment.

**AT: l357 " indebted" : i suggest "due"**

Reply: Will be fixed.

**AT: l398 I agree with your sentiment that cooperative efforts are needed to document coevolution. If possible, can you share any thoughts what that could look like? I can imagine that thermochronology, for instance, can help constrain the topographic side of the co-evolution. Is the problem mostly with reconstructing spatiotemporal patterns of rainfall?**

Reply: Thanks for thoughtful comment. Encouraged by AT, we will add a sentence "On this issue, we indeed witness new lights such as thermochronology in topography reconstruction and paleo-climate simulations using General Circulation Models."

――――――――――――――――

[Figure]

Fig. 1.

---

## Author Comment (AC2) · 3 Feb 2021

We thank Referee 2 for positive review comments. In particular, we appreciate detailed editorial suggestions given as an annotated pdf file. We will accommodate all these suggestions in the revised manuscript which is allowed to be submitted by HESS in the next step. Our replies to two questions of Referee 2 are listed below.

**[Referee 2] line 184-185: why would over-erosion happen, please elaborate and explain in more depth**

Reply: We will add a sentence as "This occasion is considered as an artifact due to the limited resolution of a numerical approach. The sediment flux is a function of the local slope (equation (13)). Let us take an example of two adjacent cells where the

upstream cell is to be eroded and the eroded material is supposed to be deposited on the downstream cell. In reality, the local slope between them gradually reduces as the erosion-deposition continues. Accordingly, the sediment flux decreases and becomes nil if the gradient reaches zero. Therefore, no further erosion is possible. However, in the discrete modeling, the local slope is invariant for the calculation time interval. Therefore, the sediment flux from the upstream cell is likely to be over-estimated. If the local slope varies greatly within the calculation time interval, the over-estimation of sediment flux and so erosion can be excessive, which can lower the cell even below the downstream cell. To prevent or minimize over-erosion, it would be necessary to reduce the calculation time interval. This can be done by dividing the storm duration into multiple time steps."

**[Referee 2] line 398: while the authors acknowledge that the current work is based on explorations on a virtual landscape, they indicate that testing their results on real landscapes is a challenge. The authors indicate they need to collaborate to obtain co-evolution data. Indeed this is an issue, but still it would be good if the authors could include some first ideas or examples of how / where their model could be tested.**

Reply: We will add a sentence as "On this issue, we indeed witness new lights such as thermochronology in topography reconstruction and paleo-climate simulations using General Circulation Models."

---

## Author Response (AR1)

**Reply for the Refree1(AT)'s comments**

We thank Arnaud Temme (referred to as AT in the following) for thoughtful review comments. In particular, we are very pleased to find that AT recognizes the value of our study. We also appreciate line-by-line comments provided, which are very helpful to further improve the quality of this paper. Our reply to each point is listed below.

**AT: l18 first mention of hydrometeors - perhaps explain this to the uninitiated (is it raindrops, or raindrops and/or snowflakes and/or hail, for instance)**

**Reply:** According to the American Meteorological Society, the 'hydrometeor' is defined as "any product of condensation or deposition of atmospheric water vapor, whether formed in the free atmosphere or at the earth's surface; also, any water particle blown by the wind from the earth's surface." We have considered writing this definition in the text but decided not to do for two reasons: (1) this is a widely known terminology and (2) writing the definition requires a separate sentence, which can pause the smooth flow of the logic of the given paragraph.

**AT: l40 rainfall depth as well is an unusual term (to me at least). Please add a few words**

**Reply:** We replaced it with 'rainfall amount.'

**AT: l69 "wonder" - do you perhaps mean "question" ?**

**Reply:** Yes, we corrected it.

**AT: l92 non-orographic large-scale vertically integrated condensation rate - so, does that mean through convectional uplift? I am not sure, but if it is, maybe that is worth mentioning to clarify the contrast with orographic uplift.**

**Reply:** For clarification, we revised the sentence as "Here, $S_o$ is the background condensation rate, driven by the large-scale (synoptic) vertical wind component. If the orographic effect is nil (either $\vec{v}$ or $\nabla{z}$ is zero), $S=S_o$" (L92-93)

**AT: l136 "the entire landscape has been assumed..." - not quite. Models by Schoorl and by Lague, for instance, deal with both conditions, as far as I understand. I placed references below.**

**Reply:** Thanks for the reference. We agree that the statement cannot be applicable for 'all' modeling studies. We revised the sentence as "In most theoretical modeling studies, however, the entire landscape has been assumed as one of these conditions." (L136)

**AT: l168 how do you deal with sediment eroding off the mountain range and entering the sea-cells? Is it assumed to disappear? The remainder of your text does suggest that, but it would be good to mention that here.**

**Reply:** We added the following sentence: "Sediments eroded over the landscape, entering the surrounding ocean, are considered permanently lost, i.e., sea floor is assumed as of an indefinite depth." (L175-176)

**AT: l184 "is of a significant amount and so zdeta" . I suggest: "and so IS zdeta"**

**Reply:** Fixed

**AT: l207 Please specify the amount of noise (perhaps by providing the ranges of the uniform distribution), and I also suggest to confirm in the text that the noise is not spatially auto-correlated.**

**Reply:** As suggested, it is revised as "We adopted the former approach and the random noise, following a uniform distribution (not spatially auto-correlated), is given in the initial topography. The initial topography is nearly flat with a very mild slope ($10^{-5}$) imposed to make sure surface water flows towards the ocean. The given random perturbation ($\pm 5 \times 10^{-4}$ m) is tiny enough relative to the given initial valley gradient, and so no depression zone forms in the initial topography." (L218-221)

**AT: l208 Please specify the initial slope value. This is especially relevant since earlier work has shown extreme sensitivity to the initial slope (in non-uplifting settings at least). I think that work used the Child model and the original may be in a paper first-authored by G.E. Tucker, but I couldn't find it anymore for you.**

**Reply:** See above reply

**AT: l211 At this point, you have explained the model in mathematical detail. As noted before, I feel that that is well done. However, I would really appreciate it if you could provide a figure about here that shows how the rainfall works out in model simulations. Perhaps showing the amount and net direction of descent of precip from the airmass moving over a hypothetical topography would work best, but any output that shows a bit of what is calculated before getting to the excess rainfall that you show from here on out, would be welcome.**

**Reply:** We generate a new Figure 2 which should address AT's request.

**AT: l242 "landslide events" I don't believe you have yet told us that you simulate landsliding - or how it is done. Are you simply using a threshold slope gradient?**

**Reply:** Yes, it is stated in L152-155 as "Shallow landslide is considered in the simple way, previously applied by \citet{Tucker1994}; any slopes greater than the angle of repose $\Theta$ shall fail, translating upslope mass at the rate $q_f$ to downhill until the local slope reaches $\Theta$. This contribution is combined with the fluvial sediment transport to form the total sediment flux $\xi (= q_s + q_f)$ at every cell."

**AT: l243 "fluctuations". That process is also discussed in Temme, 2006. I provide the reference below, feel free to disregard.**

**Reply:** Yes, the variation of SDR in Temme (2006) is similar to what we see in this study. The reference is cited.

**AT: l262-264 I felt that this information is perhaps better placed in the model description section, and is distracting here.**

**Reply:** We concur and relocated this part to section 2 (L85-87).

**AT:** Figure 3: The legend font is too small here. I noticed that the font size varies substantially among figures. Can you choose one font size and apply it to all figures?

**Reply:** Corrected

**AT:** Figure 3 caption: Numbers in the legend indicate (no "s")

**Reply:** Corrected

**AT:** l305-310 Can you here compare with overall topographic features in some of the world's mountain ranges that are closest to your condition that U=0? I believe the Andes have U=0, approximately. Do the Andes (or other mountains) match your simulated topography with feedbacks better than they do the topography without feedbacks, or even better than the topography with feedbacks from the Roe at al model? If this is not possible, you might consider here pointing at the difficulty of making comparisons with existing orogens.

**Reply:** We appreciate this insightful comment. The Andes could be a great target to be examined. We, however, determine that such comparison with real topography is premature at the current stage, although this could be a great topic for future study without a doubt. There are still several technical improvements needed from the numerical model side as we stated in the limitation section. We also added the role of vegetation in that section as "The scope of the present study is the bare soil landscape. In reality, vegetation has played a significant role in shaping the Earth's terrestrial surface. Incorporating vegetation dynamics could make co-evolution results much more complicated, as their feedback mechanisms have complex link with water, solar radiation, nutrient, carbon, sediment (soil), topography, etc. Modeling vegetation dynamics in the whole landscape context has gradually progressed in the last decade \citep[e.g.,][]{McGrath2012,Yetemen2015}. These efforts promise that vegetation dynamics can be eventually incorporated into the proposed co-evolution framework." (L410-415)

**AT:** l327 I suggest " was found to be nonlinear"

**Reply:** Fixed

**AT:** l344 Opposite: I sugggest "In contrast"

**Reply:** Fixed

**AT:** l345/6 Visualizing this rainfall displacement through wind and delayed formation is exactly what would work really well in a "Figure 0" as mentioned above.

**Reply:** See above reply. The new Figure 2 would address this comment.

**AT:** l357 " indebted" : i suggest "due"

**Reply:** Fixed

**AT:** l398 I agree with your sentiment that cooperative efforts are needed to document coevolution. If possible, can you share any thoughts what that could look like? I can

**imagine that thermochronology, for instance, can help constrain the topographic side of the co-evolution. Is the problem mostly with reconstructing spatiotemporal patterns of rainfall?**

**Reply:** Thanks for thoughtful comment. Encouraged by AT, we added a new sentence: "On this issue, we indeed witness new lights such as thermochronology in topography reconstruction and paleo-climate simulations using General Circulation Models." (L417-419)

**Reply for the Refree2(R2)'s comments**

We thank Referee 2 for positive review comments. In particular, we appreciate detailed editorial suggestions given as an annotated pdf file. We accommodated all these suggestions in the revised manuscript. Our replies to two questions of Referee 2 are given below.

**R2: line 184-185: why would over-erosion happen, please elaborate and explain in more depth**

**Reply**: We added detailed explanation as "This occasion is considered as an artifact due to the limited resolution of a numerical approach. Let us take an example of two adjacent cells where the upstream cell is to be eroded and the eroded material is supposed to be deposited on the downstream cell. As the erosion-deposition continues the local slope between them gradually reduces. Because the sediment flux is a function of the local slope (equation (\ref{eq:q_s})), the sediment flux accordingly decreases and becomes nil if the gradient reaches zero. However, in the discrete modeling, the local slope is invariant for the calculation time interval. Therefore, the sediment flux from the upstream cell is likely to be over-estimated. If the over-estimation is excessive, the elevation of the cell can become even below the downstream cell. To prevent or minimize over-erosion, it would be necessary to reduce the calculation time interval. This can be done by dividing the storm duration into multiple time steps." (L191-198)

**R2: line 398: while the authors acknowledge that the current work is based on explorations on a virtual landscape, they indicate that testing their results on real landscapes is a challenge. The authors indicate they need to collaborate to obtain co-evolution data. Indeed this is an issue, but still it would be good if the authors could include some first ideas or examples of how / where their model could be tested.**

**Reply:** We added a new sentence: "On this issue, we indeed witness new lights such as thermochronology in topography reconstruction and paleo-climate simulations using General Circulation Models." (L417-419)

**Reply for the Refree3(OY)'s comments**

We thank Omer Yetemen (referred to as OY in the following) for constructive review comments. In particular, we are very pleased to find that OY recognizes the value of our study. We also appreciate detailed line-by-line comments provided, which are very helpful to further improve the quality of this paper. Our reply to each point is listed below.

**OY: L172-173. "Whole landscape evolution modeling usually runs over the geologic time scale. Accordingly, calculation time interval is usually greater than a year. On the other hand, a storm event lasts at best a few days." Landscape evolution models not necessarily work on annual time scale. Currently, there are modelling examples run on hourly precipitation and daily precipitation intervals (e.g., Yetemen et al. 2015).}**

**Reply:** We agree with OY, and so we mentioned 'usually.' The purpose of these sentences is to explain our approach to the majority of readers who would be familiar with typical LEMs with a long time interval.

**OY: L196-197. "For the purpose of this study, an ideal simulation target can be a high and active mountain range (such as Andes and Himalaya) where orographic effect appears clearly." Orographically enhanced precipitation can be more obvious in these mountain range; however, it can be effective in intracontinental semiarid locations too (Wainwright, 2005). Wainwright, J. (2005). Climate and climatological variations in the Jornada experimental range and neighbouring areas of the US southwest. Advanced Environmental Monitoring and Modelling, 2, 39–110.**

**Reply:** Thank you for good information and reference. This supports our modeling setup.

**OY: L208-210. "The initial topography is nearly flat with a very mild slope imposed to make sure surface water flows toward the ocean. The given random perturbation is tiny enough relative to the given initial valley gradient, and so no depression zone forms in the initial topography." A bit more clarification is needed!**

**Reply:** As suggested, it is revised for clarification as "The initial topography is nearly flat with a very mild slope ($10^{-5}$) imposed to make sure surface water flows towards the ocean. The given random perturbation ($\pm 5 \times 10^{-4}$ m) is tiny enough relative to the given initial valley gradient, and so no depression zone forms in the initial topography." (L219-221)

**OY: What is the direction of flow, D8-steepest descent?**

**Reply:** Flow direction algorithm is stated in L129-134 as "LEGS adopted the GD8 method for surface flow path extraction. While this helps obtain reasonable results in terms of evolutionary speed and characteristics of simulated topography, some technical issues have been reported with the GD8 method, which are resolved by the improved GD8 method. The improved GD8 method requires the assignment of a start cell in a digital elevation model. This requires sorting cells according to their elevation values. Merge sort algorithm, invented by John Von Neumann \citep[see e.g.,][]{Knuth1987}, is adopted for this task."

**OY:** So, the perturbation must ensure the some of the side slopes of the cells must be steeper than those of the profile slope. Otherwise, all flow drains straight downward, and you cannot generate meandering or dendritic channels!

**Reply:** The perturbation and profile slope (stated in the above reply) in our simulations are carefully determined to generate dendritic networks.

**OY: L213. "No-feedback simulation results are obtained by running the LEGS model with spatially uniform rainfall, invariant over time." Do the authors mean that a single precipitation pulse is repeating during the entire simulation?**

**Reply:** Yes, please refer to our rainfall scenario description in the main text.

**OY: L277. For Figure 2b, can the authors provide/add 3D (oblique) view of the landscape? I am very curious to see this simulated topography. Following figure, 3b, 2D figure shows the densification in the counter lines but too black and height is not clear! The mean elevation profile of Figure 2b is extremely asymmetric! Huge cliff on the leeward side, just about 5 km length. I am very surprised to see such asymmetric behavior when I compare with previous papers (Goren et al. 2014; Han et al., 2014; Han et al., 2015). Is it artifact of the model? Goren et al. 2014. Coupled numerical–analytical approach to landscape evolution modeling. Earth Surf. Process. Landforms 39, 522–545.**

**Reply:** Upon the request, we generated a new Figure 2, which shows 3D view of the landscape. We believe that a new figure would satisfy OY's curiosity. Please note that asymmetry can greatly vary depending on topographic and climatic settings.

**OY: L296-7. "This leads to the downward peak migration and a steeper slope on the lee side (Figure 4c)". This definition can be valid for bare soil (as seen in lab experiments of Bonnet and Crave, 2003); however, in vegetated environment this may not be correct. Enhanced vegetation on the windward side as a result of more rainfall may impede erosion. The steepest side is defined by the competition between the erosion inhibition by vegetation and enhanced erosive power of increasing rainfall!**

**Reply:** We concur. Please note that the role of vegetation is beyond the scope of this study. In fact, this will be the subject of our forthcoming research. In recognition of this important point, we added following texts near the end of section 4: "The scope of the present study is the bare soil landscape. In reality, vegetation has played a significant role in shaping the Earth's terrestrial surface. Incorporating vegetation dynamics could make co-evolution results much more complicated, as their feedback mechanisms have complex link with water, solar radiation, nutrient, carbon, sediment (soil), topography, etc. Modeling vegetation dynamics in the whole landscape context has gradually progressed in the last decade \citep[e.g.,][]{McGrath2012,Yetemen2015}. These efforts promise that vegetation dynamics can be eventually incorporated into the proposed co-evolution framework." (L410-415)